# Positive Psychology Approaches to Interventions for Cancer Dyads: A Scoping Review

**DOI:** 10.3390/ijerph192013561

**Published:** 2022-10-19

**Authors:** Amy K. Otto, Dana Ketcher, Maija Reblin, Alexandra L. Terrill

**Affiliations:** 1University of Minnesota Medical School, Duluth Campus, Duluth, MN 55812, USA; 2College of Medicine, University of Vermont, Burlington, VT 05405, USA; 3Department of Occupational & Recreational Therapies, University of Utah, Salt Lake City, UT 84112, USA

**Keywords:** positive psychology, dyads, cancer, cancer survivors, caregivers, interventions

## Abstract

Objective: Positive psychology approaches (PPAs) to interventions focus on developing positive cognitions, emotions, and behavior. Benefits of these interventions may be compounded when delivered to interdependent dyads. However, dyadic interventions involving PPAs are relatively new in the cancer context. This scoping review aimed to provide an overview of the available research evidence for use of dyadic PPA-based interventions in cancer and identify gaps in this literature. Methods: Following PRISMA guidelines, we conducted a scoping review of intervention studies that included PPAs delivered to both members of an adult dyad including a cancer patient and support person (e.g., family caregiver, intimate partner). Results: Forty-eight studies, including 39 primary analyses and 28 unique interventions, were included. Most often (53.8%), the support person in the dyad was broadly defined as a “caregiver”; the most frequent specifically-defined role was spouse (41.0%). PPAs (e.g., meaning making) were often paired with other intervention components (e.g., education). Outcomes were mostly individual well-being or dyadic coping/adjustment. Conclusions: Wide variability exists in PPA type/function and their targeted outcomes. More work is needed to refine the definition/terminology and understand specific mechanisms of positive psychology approaches.

## 1. Introduction

Nearly 40% of men and women will be diagnosed with cancer during their lifetimes [1]. Many of these individuals receive substantial, unpaid, support from an informal caregiver, often a family member or friend. Both patient and caregiver psychological and physical health have been shown to be impacted by cancer, and importantly, many studies have documented the interdependence of psychological and physical outcomes in patients and their caregivers [2,3]. Traditionally, psychosocial and behavioral interventions to improve patient and caregiver health have been targeted toward the individual [4,5]. However, because of the interdependence between patients and caregivers [6], research has espoused the benefits of dyadic behavioral interventions to support well-being. In these interventions, two people—often the person with cancer and their caregiver—are active participants in the intervention. Findings suggest the dyadic approach is effective for improving well-being, including depression, anxiety, and quality of life [7,8].

Many existing psychosocial interventions, including those developed for dyads, are pathology- and deficit-oriented. Positive psychology offers a re-orientation to this approach, as it is a field of psychological theory and research that focuses on the psychological states, individual traits, and social institutions that enhance subjective well-being [9]. As such, interventions that are based on or incorporate aspects of positive psychology (from here on referred to as positive psychology approaches, or PPAs) aim to supplement the traditional “fix-what’s-wrong” model and seek to build on individuals’ strengths, resources, and values to increase overall well-being [10]. Importantly, PPAs do not deny or ignore the negative; rather, they aim to provide a more balanced approach to treatment to develop positive cognitions, emotions, and behaviors [11]. Specific intervention activities that are part of PPAs vary but can be generally grouped by the five pillars of positive psychology: enhancing positive emotions, engagement, positive relationships, meaning-making, and accomplishment (PERMA) [12,13]. In general, PPAs have few to no negative side effects and require comparably fewer resources than traditional therapies or interventions [14,15]. These activities can often be delivered in person one-on-one, in groups, by phone, and/or online or in self-guided formats, and may be the sole focus or one element of a multi-component intervention.

PPAs have been effectively applied to various populations [16,17,18], and have been shown to significantly increase well-being and decrease depressive symptoms [10,11,14] with long-lasting effects [19]. In cancer populations, a number of studies based on PPAs have demonstrated positive outcomes for individuals. For example, meaning-centered group therapy was shown to be effective for cancer survivors to improve personal meaning, psychological well-being and adjustment to cancer in the short term, and over long term, reduce psychological distress [20]. An online gratitude intervention was found to decrease death-related fear of recurrence in breast cancer patients [21]. Additionally, a systematic review of positive psychology interventions in breast cancer found positive changes in breast cancer patients’ quality of life, well-being, hope, benefit finding, and optimism [22]. 

PPAs are typically targeted at individuals but are well-suited for dyads. They may promote individual benefits, but there may also be synergistic benefits due to interdependence of well-being and quality of life outcomes in close dyads [6,23]—that is, as one partner experiences improvements in mood/stress, this may positively benefit the other partner’s mood. For example, a study examining savoring (i.e., purposively attending to past, present, and potential future positive experiences to enhance positive cognitions and emotions [24]) in family dyads coping with cancer found that, in addition to savoring being associated with one’s own positive affect and life satisfaction, the patient’s savoring was associated with the caregiver’s positive affect, and caregiver savoring was associated with the patient’s life satisfaction [24].

Although there is growing evidence that dyadic PPAs may be useful to improve key psychosocial outcomes in cancer patients and caregivers, the literature on dyadic PPAs in cancer populations is newer and less well-established. As such, the objective of this scoping review was to conduct a thorough search of the literature to provide an overview of the available research evidence for use of a dyadic PPA with cancer patients and their caregivers.

## 2. Materials and Methods

### 2.1. Definition of PPA and Scope of Review

Because positive psychology has not historically focused on a single, refined approach, the field’s scope is large and there is substantial heterogeneity [25]. However, we use theory and previous reviews of PPA research (e.g., [14,22]) to guide our definition of PPA and the scope of this review. A PPA was broadly defined as a psychological intervention or therapeutic approach that primarily focused on building on existing strengths and resources—both personal and interpersonal—to meet life’s challenges and actively facilitate growth, resilience, and well-being [10,26]. 

Our scope included interventions rooted in positive psychology theory or tradition (e.g., well-being therapy, hope therapy) and interventions whose primary goal is to increase positive feelings, positive cognitions, or positive behavior, as opposed to interventions aiming to reduce symptoms, problems, or disorders. Interventions focused primarily on one or more of the five pillars of positive psychology noted earlier were included; for example, interventions that emphasize focusing on the positive or that enhance the enjoyment of positive experience (savoring), and interventions that promote meaning and purpose. Interventions that are primarily mindfulness-based (e.g., mindfulness-based stress reduction/MBSR) or interventions that include mindfulness along with other PPA-based components were included, as mindfulness is linked to the positive psychology pillar of positive emotion, and mindfulness is often considered a tool of positive psychology [27].

We excluded studies that focused primarily on activities-based interventions that are not explicitly rooted in positive psychology, including yoga or other physical exercises for purposes of promoting well-being, relaxation and imagery/visualization exercises, art therapy, and music therapy. We also excluded interventions that do not have a primary focus on at least one of the five pillars of positive psychology, including behavioral activation, psychoeducational interventions, cognitive behavioral therapy, cognitive therapy, cognitive behavioral stress management, acceptance and commitment therapy, dialectical behavior therapy, problem-focused therapies, psychodynamic therapy, and supportive-expressive therapy.

### 2.2. Information Sources and Search Strategy

Our team developed a list of search terms (See Appendix A) based on our definition of PPAs. A detailed and systematic search was conducted between May 2019 and September 2021 using a combination of free-text-keywords, MeSH, and database-specific controlled vocabulary within PubMed.gov, EMBASE, and Central (Cochrane Library). Search results were downloaded in RIS format from each database/website and imported into an EndNote library. Once compiled into the library, search results were deduplicated three times, once using EndNote, once upon uploading into Covidence (Veritas Health Innovation), and finally once during the title and abstract screening process. Backward searches were conducted using citations in reviews and meta-analyses identified in our initial search until no additional relevant articles were found. Two reviewers were used at each stage of screening (i.e., title and abstract screening, full-text screening).

### 2.3. Eligibility Criteria

See Appendix A for a detailed description of inclusion and exclusion criteria. We searched for studies related to PPAs, as described above, delivered to adult dyads comprised of a cancer patient/survivor and one member of the survivor’s informal social support network (i.e., family member or friend). We stipulated that the intervention must be delivered to both members of the dyad together for at least part of the intervention, although the intervention did not have to place equal emphasis on both dyad members. For example, the intervention could be delivered to patients individually for most sessions and to the dyad together for a smaller portion of the sessions, and/or outcomes did not have to be assessed or analyzed for both dyad members (e.g., analyses could focus exclusively on patient outcomes). (See the Discussion section below for additional discussion on the variety of ways that “dyadic” may be interpreted in this context.) 

English-language peer-reviewed articles reporting results of original research—e.g., pilot trials, randomized control trials, and secondary analyses—were included. Articles were not excluded based on publication date. Articles which merely described a study protocol or intervention and did not report results were excluded, such as published and unpublished protocols, methodology write-ups, and guideline reports. Case studies and non-peer-reviewed theses, dissertations, and book chapters were also excluded. Meta-analyses and reviews were not included, but their reference lists were hand-searched to find relevant articles that may have been missed in our database searches.

### 2.4. Study Selection Process

The screening and review process followed Preferred Reporting Items for Systematic Reviews and Meta-Analyses (PRISMA) guidelines (http://www.prisma-statement.org/, accessed on 14 January 2022). See Figure 1 for the PRISMA flow diagram. A total of 3316 articles were identified through the database search, with an additional 10 articles identified via hand search (3326 total). From these, 232 duplicates were removed, resulting in 3094 studies screened by authors against the inclusion/exclusion criteria. A total of 2868 articles were excluded during the title/abstract screening process. A total of 226 articles were moved to full-text review; 180 were excluded, most commonly because they did not include a PPA (*n* = 62, 34.4%) or they were not original research (e.g., reviews, meta-analyses; *n* = 58, 32.2%). Ultimately, 48 studies were selected for data extraction and reporting. Consistent with scoping review guidelines, methodological quality and risk of bias were not assessed [28].

## 3. Results

Search results yielded 48 articles, all published between 2002–2021. See Table 1 for a summary; Appendix A shows demographic information for each study’s sample. Twenty-eight unique interventions were assessed across these 48 articles; 21 of these 28 interventions (75.0%) were reported in only a single publication. Thirty-nine of the 48 articles were primary analyses; eight articles (16.7%) were secondary analyses [29,30,31,32,33,34,35,36], and patient and caregiver results for one trial were reported in two separate articles [37,38] and were counted together as a single primary analysis. Primary studies included a variety of cancer sites, including breast (*n* = 7 of 39, 17.9%), prostate (*n* = 6, 15.4%), lung (*n* = 6, 15.4%), and mixed or unspecified site (*n* = 16, 41.0%). Overall, studies encompassed a wide range of cancer stages, from early stage to hospice care; five primary studies (12.8%) specifically focused on advanced cancer. Most primary studies were randomized controlled trials (*n* = 23 of 39, 59.0%), and 14 were single-arm trials (35.9%). There was substantial variability in sample size across studies, ranging from 5–484 dyads. Most studies (*n* = 37 of 48, 77.1%) were published after 2010; 12 (25.0%) were published in 2019 or later. Most primary studies were conducted in North America (*n* = 31 of 39, 79.5%).

The FOCUS intervention was published on by far the most frequently, with 11 studies (22.9% of 48 included studies, including three secondary analyses) assessing variations of the intervention: brief/extended versions, self-managed web-based versions, and a group-based version. COPE and DYP were the only other interventions included in more than two papers. 

There was substantial variation in how the non-patient dyad member was defined. In 21 of the 39 primary studies (53.8%) this dyad member was classified under a general “caregiver” label, which was defined in a variety of ways and sometimes not at all; however, by far the most common type of caregiver was the patient’s spouse/partner, with an average of 71% of participating caregivers being the patient’s spouse/partner, across studies that reported this information. Study inclusion criteria for caregiver participants varied by time (e.g., time living together, times visited patient), relationship type (e.g., friend, family), or by who “provided the most care” to the patient. Sixteen primary studies (41.0%) specifically recruited patients’ spouses/partners (vs. “caregivers” defined more broadly), which varied in including/excluding same-sex partners. The studies that specifically enrolled spouses/partners almost exclusively dealt with breast and prostate cancer.

### 3.1. Positive Psychology Approaches: Intervention Content

Table 2 contains a summary of each intervention’s structure and content, as well as relevant pillar(s) of positive psychology. Although the term “positive psychology” was explicitly mentioned in relation to only one intervention in this review (CBM [41]) aspects of positive psychology were found throughout the descriptions of interventions: Each intervention included a primary focus on activities relevant to at least one and up to four of the five positive psychology pillars. Most commonly, interventions included components related to the positive emotions pillar (*n* = 20 of 28 unique interventions, 71.4%); in this review, we further categorized positive emotion components into those focused on mindfulness, present in 10 interventions (38.5%), optimism/hope, and other/general positive emotions, both in six interventions (23.1%). Meaning-making and positive relationships were also commonly-represented pillars, each in 11 interventions (39.3%). Activities related to engagement and accomplishment pillars were less common, each in two interventions (7.7%). Beyond CBM [41], which was explicitly informed by positive psychology processes, no intervention was exclusively composed of PPAs. Rather, aspects of positive psychology were included with other dyadic intervention components such as education, managing symptoms, coping, and intimacy. 

### 3.2. Outcomes Assessed

Given the relative novelty of dyadic interventions in cancer using PPAs, it is not surprising that many studies focused on feasibility, acceptability, or satisfaction of study components (*n* = 19 of 48; 39.6%) as primary outcomes. However, a broad range of psychosocial constructs were also assessed as primary outcomes (see Table 1). The most common outcomes were quality of life (*n* = 27 of 48; 56.3%), depression (*n* = 14; 29.2%), and anxiety (*n* = 10; 20.8%), though a variety of unique measures were used to assess these constructs. Overall patterns of results suggest trends towards intervention effectiveness in improving quality of life and reducing depression and anxiety in at least one member of the dyad. Other constructs were too infrequently assessed to make generalities about effectiveness.

## 4. Discussion

This scoping review highlights the current research on positive psychology approaches (PPAs) in dyads coping with cancer. Findings show that dyadic interventions using PPAs are increasingly used in oncology, particularly as part of multi-component interventions. Interventions were delivered over multiple sessions by an interventionist either to individual dyads or to dyads within a group context. Interventions targeted a wide variety of cancer patients, across both disease site and trajectory, and the type of dyad partner also varied; while many dyads included the patient’s spouse or partner, most studies included a broadly-defined “caregiver.” The primary outcomes assessed in these interventions were individual psychological well-being and quality of life or distress (including anxiety and depression), dyadic coping or adjustment, and, less often, physical symptoms such as fatigue or pain. This spectrum of intervention participants and targeted outcomes speaks to the flexibility and potential broad application of PPAs, especially as part of a multicomponent intervention.

This review found a variety of activities relevant to positive psychology were used in interventions. Activities were most commonly focused on pillars of positive emotion—especially mindfulness, and optimism or hope—as well as meaning-making and positive relationships. These constructs are particularly well-suited for dyadic interventions in oncology. First, mindfulness, hope, and meaning-making can all be helpful coping tools for cancer patients and their family caregivers [77,78,79,80,81,82]; these cognitive strategies may be especially useful in oncology settings where individuals may feel they have little control over the cancer. Dispositional optimism also has important impacts on symptom experience and quality of life for cancer patients [83] and all-cause mortality more broadly [84]. Cancer has long been viewed as a “family disease”; patients often include family members in decision-making and increasingly rely on caregivers, especially as their health declines [85,86]. This, combined with the benefits of social support in cancer [87,88], means that the oncology setting may especially lend itself to dyadic interventions and a focus on positive relationships.

While the breadth of activities within PPAs demonstrates the wide applicability, similar to previous reviews [14,22], we find this can also create difficulties in pinpointing specific benefits. Since many existing interventions include PPAs as one component among several others (e.g., education, problem-solving, social support), it is difficult to evaluate what may be driving the effects of a given intervention. Some PPAs may also serve as multipliers for the effects of more traditional intervention tools. For example, mindfulness exercises may help participants to focus, reducing anxiety and making problem-solving or education more effective. Finding meaning in their experience may give dyads a deeper well of resilience to draw from when coping with the stresses of cancer. However, particularly because most interventions identified in our search were multi-component, more work is needed to identify mechanisms and determine which specific PPAs are most effective for whom and in which situations. Similarly, when multiple PPAs are employed in a study, it may be beneficial to disentangle how those work together in different contexts to affect key outcomes.

Similarly, other researchers have called for more work to better understand how, why, and for whom dyadic interventions specifically are effective [7]. PPAs typically target the individual, and much of the empirical research supporting the effectiveness of PPAs is based on individual participants, not dyads. It is undeniable that the dyadic aspect of an intervention adds another dimension, which should also be considered in identifying mechanisms. The high level of interdependence of mental and physical health between cancer patients and their family caregivers or spouses, who often participate in dyadic studies, may mean that independent effects are compounded, plus there may be additional unique dyadic effects [6,23]. For example, a gratitude activity between partners may enhance positive emotion and feelings of connectedness in the dyad, which could promote coping with cancer-related stress. 

Though many of the studies included in this review focused (appropriately) on feasibility and acceptability, there remains a need to identify which outcomes are most appropriate to measure as key intervention targets. Our review suggests that dyadic interventions with PPAs may be beneficial in terms of quality of life, depression, and anxiety, yet there is no consensus around other constructs that may be impacted, or how to measure these constructs. More work mapping out mechanisms and outcomes to a conceptual model and testing these theories can be beneficial in moving the field forward. Further, using validated measures that allow for data harmonization will be important for future meta-analyses.

Although all interventions in this review included dyads, not all interventions were purely dyadic. In several interventions, some sessions were targeted specifically for patients and included both patient and their dyad partner only in select sessions. For example, the QOL intervention [30] held six sessions for patients, and caregivers were invited to four of these sessions deemed most relevant to both patients and caregivers. Other interventions were delivered to both dyad members, but either due to the specific activities selected or the design of the intervention, maintained an individual focus. Examples include the TSM intervention [34,71], in which both the patient and partner were coached in relaxation exercises, and the COPE intervention [32,45,46], in which both patient and partner were provided guidance in maintaining an optimistic outlook for themselves. This contrasts with other, more truly dyadic, interventions in which dyads participate in activities together and the focus is more on the *interaction* or *interdependence* of activities. For example, the ECG intervention [52] included a “wish list” of positive acts each spouse could do for the other and leveraged support to make changes and increase intimacy. Similarly, the FOCUS intervention [55,57,59] included brainstorming positive activities to do together and relies on family strengths. Many interventions also included a hybrid model of both individual and dyadic targets, which may be an ideal strategy to leverage the benefits of interdependence. Additional research is needed to identify benefits to different levels of dyadic inclusion across different activities and interventions.

### 4.1. Limitations and Future Directions

It is well-documented that recruitment and retention of dyads in cancer research is challenging, particularly in advanced cancer contexts. Interventions in our review targeted dyads across the cancer trajectory from diagnosis to end of life. While interventions were largely feasible and offered benefits across this trajectory, most studies in our review were relatively small, and most of their findings have not yet been replicated. Further, the majority of studies sampled patients and caregivers who were mostly White, relatively high-socioeconomic status, and from the US or other Western countries, and spouses/partners were by far the most common type of dyad partner (vs. adult children, siblings, etc.) represented in these studies (see Table 1 and Appendix A). It is unclear how selection bias may impact the uptake and effectiveness of these interventions. Future research with larger, more diverse samples is needed. 

Additionally, although (by design) all interventions in this review targeted dyads—a cancer patient and a partner—the specific role or relationship of the partner may vary (and in some cases, was not clearly defined). For example, some studies focused specifically on romantic partners, while others focused on primary caregivers. These two roles may be taken on by the same individual (i.e., a romantic partner might also be the primary caregiver), but there is certainly some variability—caregivers may also be adult children, siblings, or others, and some romantic partners may also not be highly involved in providing care for the cancer patient. This variability needs additional exploration.

Finally, our review may be limited by a lack of consistent terminology and a consensus definition of “positive psychology approaches” or “positive psychology interventions” [14,22]. Only one intervention was explicitly based in positive psychology, though all included PPAs. This lack of specificity may arise from the relatively recent developmental history of positive psychology [89], which was born from Martin Seligman’s theme during his American Psychological Association presidency to more broadly consider the full human experience as being both negative and positive, and to therefore encourage the latter (as well as consider the former). This called for the integration of positive psychology into existing practice, as opposed to establishing a singular approach. As such, there is not yet a clear, widely-accepted set of specific key words to identify these studies, and some dyadic PPAs may not have been identified in this scoping review. We found PPAs in interventions developed across disciplines, including nursing, social work, and palliative care. This demonstrates the broad appeal of these tools, yet also may contribute to the challenges of finding a shared terminology across fields of study. 

A multidisciplinary consensus group to begin creating more solid definitions and key terms may be an important step for the growth of this field. Based on our findings—as well as previously-conducted systematic reviews on positive psychology interventions—we propose the following definition: Positive psychology approaches include any intervention that contains in part or wholly aspects of positive psychology theory (e.g., PERMA) to build on recipients’ strengths, resources, and values, and whose primary goal is to increase positive feelings, cognitions, and/or behavior, which may be the sole goal of the intervention or in addition to more traditional symptom amelioration. 

### 4.2. Clinical Implications

Identifying mechanisms and key components to interventions is important for future dissemination and implementation research. Most interventions in this scoping review involved multiple sessions and many were delivered by highly-trained interventionists, such as nurses, social workers, and clinical psychologists. Understanding the specific components driving effects can help reduce resources needed to deliver interventions and may facilitate translation to alternative delivery systems (e.g., virtual or mHealth). Research supports that mHealth-delivered PPAs can be effectively applied to increase well-being and decrease depression across various populations [90,91]. Several interventions identified in our review included a phone or web-based delivery method (e.g., [65,75]), which can facilitate access to a population that is remotely located, may have mobility or transportation issues, or who simply do not have time or energy to convene for programs. 

Clinicians should note that positive psychology offers useful, relatively simple approaches to improving quality of life that can be implemented alongside other approaches, such as education. Given the feasibility of dyadic PPAs, as well as the potential for synergistic effects on well-being, clinicians can also consider including caregivers or other supportive individuals in psychosocial assessments and when offering psychosocial resources, including those that include PPAs. 

## 5. Conclusions

Positive psychology approaches hold promise to have large impacts on improving psychosocial outcomes for those coping with cancer. Further, dyadic PPAs can offer benefits to individuals, but these benefits may be compounded within dyads due to interdependence effects and partner influence. Given the high levels of anxiety and depression that are often reported in both patients and partners coping with cancer, dyadic PPAs may be important tools to improve quality of life. However, to date, the types of positive psychology-based activities that have been delivered in dyadic interventions are highly variable. More work is needed to develop terminology and understand specific mechanisms to develop this area of research and fully appreciate the potential benefits of these tools.

## Figures and Tables

**Figure 1 ijerph-19-13561-f001:**
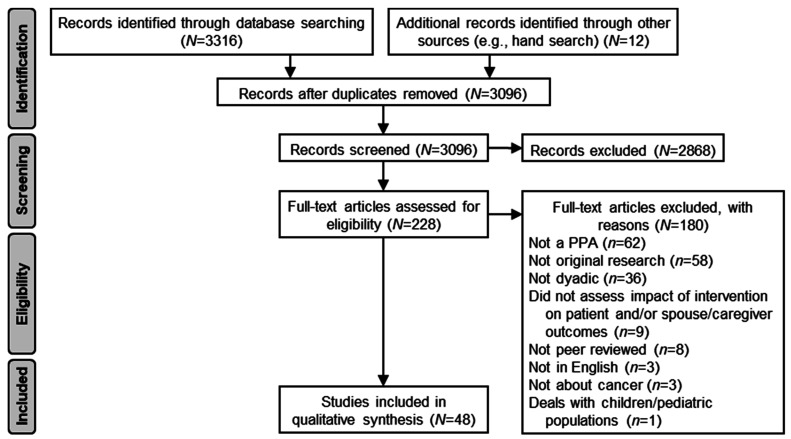
PRISMA Flow Diagram.

**Table 1 ijerph-19-13561-t001:** Summary of Studies Included in this Review.

Intervention Name	Citation	Study Type	*N* Dyads in Final Sample	Sample Description ^a^ [Location]	Primary Outcome(s)		Summary of Results
4Cs program (Caring for Couples Coping with Cancer)	Li et al. (2015) [39]	Single-arm trial	92	Cancer PTs & SPs [Hong Kong]	Self-efficacy	CBI-B	Significant improvements were seen in couples’ self-efficacy, communication, dyadic coping, physical health subscale (PCS) of MOS-SF-12, anxiety, and benefit finding; SPs had higher self-efficacy, PCS score, and anxiety than PTs
					Communication	CRCP	
					Dyadic coping	DCI	
					QOL	MOS-SF-12	
					Anxiety & depression	HADS	
					Benefit-finding	BFS	
					Relationship satisfaction	DAS	
ACT (Acceptance and Commitment Therapy)	Mosher et al. (2019) [40]	RCT: ACT vs. education/ support control condition	50	PTs with advanced lung cancer (III-IV NSCLC or extensive stage small cell lung cancer) diagnosed ≥3 weeks prior & distressed CGs (T-score ≥55 on PROMIS anxiety or depression measure, or DT score ≥3) (72% SPs) [USA]	Global symptom interference	MDASI (Global Symptom Interference subscale)	ACT did not reduce PT symptom interference or PT or CG distress
					Fatigue interference	FSI (Fatigue Interference subscale)	
					Pain interference	PROMIS-SFv1.0-Pain Interference 4a	
					Task avoidance due to dyspnea	Single PROMIS item	
					Anxiety	PROMIS-SFv1.0-Anxiety 4a	
					Depression	PROMIS-SFv1.0-Depression 4a	
					Psychological distress	Single-item DT	
CBM (Couple-Based Meditation)	Milbury et al. (2020) [41]	RCT: CBM vs. usual care	35	Brain cancer PTs & SPs [USA]	Feasibility & acceptability	Consent rates, attrition, attendance, homework completion, satisfaction	CBM was feasible, acceptable, and possibly efficacious; both PTs & SPs rated CBM as beneficial, but significant group differences (CBM vs. usual care) were only found for PTs
					Cancer-specific symptoms	MDASI-BT	
					Depression	CES-D	
					Mindfulness	MAAS	
					Intimacy	PAIRI	
CDGI (Cancer Dyads Group Intervention)	Saita et al. (2016) [42]	RCT: CDGI vs. usual care	50	Cancer PTs within 3 months of diagnosis & SPs, family members, or friends (75% SPs) [Italy]	Cancer-specific coping strategies	Mini-MAC	CDGI dyads reported increased Fighting Spirit & Avoidance and decreased Fatalism & Anxious Preoccupation coping styles, while control dyads reported increased Hopelessness/Helplessness & decreased Fatalism
					Intimacy	IOS	
CECT (Cognitive Existential Couple Therapy)	Collins et al. (2013) [43]	Single-arm pilot feasibility & acceptability trial	12	PTs with recently-diagnosed early-stage prostate cancer & SPs [Australia]	Feasibility	Participant retention	CECT was both feasible & acceptable to dyads
					Acceptability	Semi-structured interview	
	Couper et al. (2015) [44]	RCT: CECT vs. usual care	62	PTs with localized prostate cancer (T1–T3, N0, M0) diagnosed in past 12 months & SPs [Australia]	Relationship function	FRI	Compared to usual care, those in the CECT group showed improved coping for PTs, decreased cancer-related distress for SPs, & improved relationship function for both PTs & SPs
COPE	McMillan et al. (2006) [45]	RCT: COPE + usual hospice care vs. usual hospice care vs. usual hospice care + three supportive visits	329	Advanced cancer PTs in hospice & family CGs (% SPs not reported) [USA]	QOL	CQOL-C	COPE improved CG overall quality of life & decreased burden related to PT symptoms and CG tasks; COPE did not affect CG mastery
					Physical symptoms	MSAS	
					Caregiving mastery	Author-designed scale	
					Caregiving burden	CDS	
	McMillan & Small (2007) [32]	Secondary analysis of data from McMillan et al. (2006) [45]	329	See McMillan et al. (2006) [45]	Pain	NRS	COPE did not affect PT QOL or intensity of pain, dyspnea, or constipation, but it did significantly improve PT symptom distress
					Dyspnea	DIS	
					Constipation	CAS	
					Symptom distress	MSAS	
					QOL	HQLI	
	Meyers et al. (2011) [46]	RCT: COPE vs. usual care	476	Advanced cancer PTs with relapsed, refractory, or recurrent solid tumors or lymphoma on phase 1–3 clinical trials & CGs (70% SPs) [USA]	QOL	COHQOL	No group differences in PT quality of life; CGs in COPE had a smaller decline in quality of life compared to controls
					Social problem solving	SPSI-R	
CSC (Couple-Based Supportive Communi-cation intervention)	Gremore et al. (2021) [47]	Pilot RCT: CSC vs. usual care	20	PTs with non-metastatic head & neck cancer receiving active treatment & SPs [USA]	Acceptability	CSQ	98% of sessions were completed, with high levels of satisfaction with the intervention; PTs and SPs in CSC had improvements in individual and relationship functioning, relative to those in usual care
					Relationship satisfaction	DAS; MSIS	
					Post-traumatic stress	IES-R	
					Depression	CES-D	
					Anxiety	PROMIS-Anxiety	
					QOL	FACT-HN (PTs); CQOL-C (CGs)	
					Fatigue	BFI	
					Pain	BPI	
Dignity Therapy	Wang et al. (2021) [48]	RCT	68	PTs with hematologic neoplasms & CGs (47.2% SPs) [China]	Hope	HHI	PTs in intervention had higher hope, spiritual well-being, and family cohesion and adaptability vs. control group; CGs in intervention had lower anxiety, depression, and higher family adaptability vs. control group
					Spiritual well-being	FACIT-SP	
					Anxiety	SAS	
					Depression	SDS	
					Family adaptability and cohesion	FACES-II	
DYP (Dyadic Yoga Program)	Milbury, et al. (2018) [49]	Single-arm pilot study	5	High-grade glioma PTs receiving ≥4 weeks radiation & family CGs (60% SPs) [USA]	Feasibility	Consent rates; session attendance; questionnaire completion; attrition; participant evaluations	Intervention was feasible and all participants perceived the program as useful and beneficial; no statistically-significant improvements for PTs or CGs, but clinically-significant improvements seen in cancer-related symptoms, sleep disturbance, depression symptoms, and mental QOL for PTs, and in mental QOL for FCGs; however, a marginally-significant increase in depression symptoms was seen in CGs
					Cancer-related symptoms	MDASI	
					Depression	CES-D	
					Fatigue	BFI	
					Sleep disturbance	PSQI	
					QOL	MOS-SF-36	
	Millbury, Li, et al. (2019) [50]	Pilot RCT	20	PTs with grade I-IV glioma to be treated with ≥20 fractions of radiotherapy & family CGs (55% SPs) [USA]	Feasibility	Consent rates, class attendance, completion of questionnaires, attrition	Found to be acceptable and well-received by participants. Supported clinically significant decrease in overall cancer-related symptom severity, specifically affective, treatment-related, mood and GI-related symptom severity. A large effect was found in reduction of caregiver depressive symptoms. Clinically significant improvements in patient/caregiver QOL.
					Cancer-related symptoms	MDAST-BT	
					Depressive symptoms	CES-D	
					Fatigue	BFI	
					QOL	SF-36	
	Millbury, Liao, et al. (2019) [51]	Pilot RCT	26	Patients with stage I-IIIB non-small cell lung or esophageal cancer undergoing at least 5 weeks of thoracic radiotherapy & family CGs (81% SPs) [USA]	Feasibility	Consent rates, class attendance, completion of questionnaires, attrition	Intervention was feasible via a priori criteria, with 80% of dyads attending all yoga sessions. Observed clinically significant improvement in patient social function, role performance, and mental health. Caregiver treatment response less pronounced.
					Patient physical function	6MWT	
					Depressive symptoms	CES-D	
					QOL	SF-36	
ECG (Enhanced Couple-Focused Group Intervention)	Manne et al. (2016) [52]	RCT: ECG vs. couples’ support group (SG)	302	Female PTs with early-stage breast cancer who received surgery in the past 12 months & SPs [USA]	Psychological well-being	MHI (Anxiety, Depression, & Well-Being subscales)	No significant differences between ECG & SG groups; dyads in both groups had improved psychological well-being & cancer distress
					Cancer-specific distress	IES	
					Relationship satisfaction	DAS	
FOCUS	Northouse et al. (2002) [53]	RCT: FOCUS + usual care vs. usual care	117	Female PTs with recurrence or progression of breast cancer in the past month & family members (64% SPs) [USA]	Acceptability	Author-designed scale	Both FOCUS & usual care participants had high satisfaction with their care, but FOCUS PTs and CGs were more satisfied in areas of their lives that were addressed by the intervention, & FOCUS PTs felt that their nurses were more understanding
	Northouse et al. (2005) [54]	RCT: FOCUS + usual care vs. usual care	134	PTs with recurrence or progression of breast cancer in past month & family CGs (62% SPs) [USA]	Illness appraisals	AIS	PTs in FOCUS had less hopelessness at 3 months (no change among CGs); PTs & CGs in FOCUS had less negative appraisals of the illness & caregiving respectively
					Caregiving appraisals	ACS	
					Uncertainty	MUIS	
					Hopelessness	BHS	
					Coping strategies	Brief COPE	
	Northouse et al. (2007) [55]	RCT: FOCUS vs. usual care	235	Prostate cancer PTs & SPs [USA]	QOL	MOS-SF-12; FACT-G	FOCUS PTs had less uncertainty and improved communication compared to controls; FOCUS SPs had improved quality of life, self-efficacy, communication, and caregiving appraisals compared to controls
					Prostate cancer-specific QOL	FACT-P; EPIC; EPIC-S	
					Caregiving appraisals	ACS	
					Uncertainty	MUIS	
					Hopelessness	BHS	
					Coping strategies	Brief COPE	
					Cancer-related self-efficacy	LCSES	
					Communication about cancer	LMISS	
					Emotional distress	OSQ	
	Harden et al. (2009) [29]	Secondary analysis of data from Northouse et al. (2007) [55]	86	See Northouse et al. (2007) [55]	Acceptability	Author-designed scale	PTs & SPs were very satisfied with FOCUS; PTs who were better-functioning before FOCUS (e.g., better quality of life, better coping) & SPs who reported more positive changes post-FOCUS were more satisfied
	Northouse et al. (2013) [56]	RCT: brief FOCUS vs. extensive FOCUS vs. usual care	302	PTs with advanced lung, colorectal, breast, or prostate cancer within 6 months of new diagnosis, progression, or change of treatment & family CGs (74% SPs) [USA]	Illness appraisals	AIS	PTs & CGs in both FOCUS groups had more positive outcomes compared to usual care; extensive FOCUS improved dyad self-efficacy, brief FOCUS improved dyad health behaviors, and both improved dyad coping & social QOL
					Caregiving appraisals	ACS	
					Hopelessness	BHS	
					Uncertainty	MUIS-B	
					Coping strategies	Brief COPE	
					Health behaviors	Author-designed scale	
					Social support	SSQ	
					Communication about cancer	LMISS	
					Cancer-related self-efficacy	LCSES	
					QOL	FACT-G	
	Northouse et al. (2014) [57]	Single-arm feasibility study	38	PTs with lung, colorectal, breast, or prostate cancer diagnosed 2–12 months prior & family CGs (68.4% SPs) [USA]	Mood disturbance	POMS-SF	Dyads had improvements in mood disturbance & QOL
					QOL	FACT-G	
	Martinez et al. (2015) [31]	Secondary analysis of data from Northouse et al. (2013) [56]	484 patients	See Northouse et al. (2013) [56]	Health care utilization	Emergency department visits, inpatient hospitalizations (abstracted from PT medical records)	No differences in health care utilization across study groups
	Dockham et al. (2016) [58]	Single-arm pilot effectiveness study	34	Cancer survivors & family CGs (91% SPs) [USA]	QOL	FACT-G	PTs & CGs had increases in physical, emotional, functional, and overall QOL
	Titler et al. (2017) [59]	Single-arm trial	36	Cancer PTs in treatment or completed treatment within past 18 months & family CGs (% SPs not reported) [USA]	Emotional distress	CSSDS	Significant improvements were observed in overall QOL, emotional and functional well-being, and emotional distress
					QOL	FACT-G	
	Chen et al. (2021) [60]	Single-arm pilot study	29	Cancer PTs recruited from an infusion center (any site, stage, time since diagnosis; life expectancy 6+ months) & family CGs (63.33% SPs) [USA]	Coping	Brief COPE	From pre- to post-intervention, PTs and CGs showed improved self-efficacy, CGs showed improved QOL, and PTs showed decreased use of substances for coping
					QOL	FACT-G (PTs); CQOL-C (CGs)	
					Self-efficacy	LCSES	
					Acceptability	Author-designed scale	
	Titler et al. (2020) [35]	Secondary analysis of Titler et al. (2017) [59]	36	See Titler et al. (2017) [59]	Satisfaction	Author-designed scale	Participants reported that the program did not duplicate services, that it helped them cope with cancer, & that they would recommend the program to others; the most beneficial aspects of the program were the group format and dyadic approach
I-BMS (Integrative Body-Mind-Spirit intervention)	Lau et al. (2020) [37]	RCT: I-BMS vs. CBT (same data as Xiu et al. (2020) [38] report on PT outcomes)	157	Lung cancer PTs age ≥21 & CGs (68.15% SPs) [Hong Kong]	QOL	FACT-G; EORTC QLQ-30; HWS	CBT led to greater reduction in emotional vulnerability vs. I-BMS; I-BMS resulted in greater increase in overall QOL and spiritual self-care, and more reduction in depression vs. CBT; PTs in both groups had improvement in physical, emotional, & spiritual QOL
					Sleep disturbance	ISI	
					Death anxiety	Death Anxiety Scale	
					Anxiety & depression	HADS; Dysfunctional Attitudes Scale	
	Xiu et al. (2020) [38]	RCT: I-BMS vs. CBT (same data as Lau et al. (2020) [37] report on CG outcomes)	157	See Lau et al. (2020) [37]	Anxiety & depression	HADS	CGs in both I-BMS and CBT had improved QOL immediately following intervention and at follow-up; insomnia improved for both groups at T1 but deteriorated at follow-up; both groups had reduced anxiety and perceived stress at follow-up
					Perceived stress	PSS	
					Sleep disturbance	ISI	
					Caregiving burden	CRA	
					QOL	CQOL-C	
MBSR (Mindfulness-Based Stress Reduction)	Birnie et al. (2010) [61]	Single-arm trial	21	Cancer PTs (any site, stage, time since diagnosis) & SPs [Canada]	Mood disturbance	POMS	PTs & SPs had decreases in mood disturbance and in muscle tension, neurological/GI, & upper respiratory subscales of the C-SOSI, and increases in mindfulness
					Stress	C-SOSI	
					Mindfulness	MAAS	
MBSR-C (MBSR for Cancer)	Lengacher et al. (2012) [62]	Single-arm pilot study	26	PTs with stage 3–4 breast, colon, lung, or prostate cancer, who had completed surgery and were receiving radiation and/or chemotherapy & family CGs (% SPs not reported) [USA]	Perceived stress	PSS	From baseline to post-intervention, PT perceived stress and anxiety improved; CGs had decreased cortisol & IL-6 from pre- to post-session at some weeks
					Depression	CES-D	
					Anxiety	STAI	
					Physical & psychological symptoms	MSAS	
					QOL	MOS-SF-36	
					Stress markers	Salivary cortisol & interleukin-6 (IL-6)	
MODEL Care (Mindfully Optimized Delivery of End-of-Life Care)	Cottingham et al. (2019) [36]	Secondary analysis of Johns et al. (2020) [63]	12	See Johns et al. (2020) [63]	Lived experience	Qualitative interviews	PTs & CGs reported the intervention (1) enhanced adaptive coping practices, (2) lowered emotional reactivity, (3) strengthened their relationship with each other, & (4) improved their communication, including communication about cancer
	Johns et al. (2020) [63]	Single-arm pilot study	13	PTs treated for stage 3B-4 solid malignancies with prognosis of <12 months (but not in hospice) & family CGs (69.2% SPs) [USA]	Feasibility & acceptability	Accrual; attendance; retention; satisfaction & perceived helpfulness	PT engagement in advanced care planning more than doubled; PT distress decreased; CG QOL and family communication improved; PTs and CGs both had reduced sleep disturbance and avoidant coping
					Advanced care planning engagement	Completion of advanced care plan; goals of care discussions with oncologist & with family	
					Family communication	ODCNF	
					QOL	MQOL (PTs); CQOL-C (CGs)	
					Avoidant coping	Mini-MAC; Brief COPE	
					Distress	PHQ-8; GAD-7	
					Sleep disturbance	PSQI	
					Fatigue interference	FSI	
PIP/MPI (Couple-Based Psychosocial Information Package and Multimedia Psychosocial Intervention)	Chien et al. (2020) [64]	RCT: PIP vs. MPI vs. control	103	Newly-diagnosed prostate cancer PTs & SPs [Taiwan]	Disease appraisals	CAHS	SPs in MPI & PIP groups had improved positive & negative affect and mental HRQOL compared to control group. PTs were satisfied with MPI.
					Prostate cancer-specific anxiety	MAX-PC	
					Positive & negative affect	PANAS	
					Relationship satisfaction	DAS (Dyadic Satisfaction subscale)	
					HRQOL	MOS-SF-12	
					Satisfaction	Author-designed scale	
PERC (Prostate Cancer Education and Resources for Couples)	Song et al. (2015) [65]	Single-arm pilot feasibility & acceptability study	22	Localized prostate cancer PTs who completed primary treatment & SPs [USA]	QOL	FACT-G	Dyads had high website use, were satisfied with the intervention, and found it helpful; PTs had improvement in physical, social, and overall QOL
					Prostate cancer-specific QOL	EPIC	
					Communication about cancer	MISS	
					Feasibility & acceptability	Recruitment & retention rates; participant website activity; semi-structured interviews	
Partners in Coping Program (PICP)	Kayser et al. (2010) [66]	RCT: PICP vs. hospital standard social work services (SSWS)	47	Non-metastatic primary breast cancer PTs within 3 months of diagnosis and currently receiving treatment & SPs [USA]	Breast cancer-specific QOL	FACT-B	No differences in QOL for PTs and SPs in the PICP group vs. the SSWS group
					QOL	QL-SP	
					Illness intrusiveness	IIRS	
Prepared Family Caregiver Problem-Solving Intervention (PSE)	Bevans et al. (2010) [67]	Single-arm pilot feasibility study	8	Allogeneic hematopoietic stem cell transplant PTs & family CGs (100% SPs) [USA]	Feasibility	Interventionist notes (participant attendance, session length, reasons for variation)	PSE was feasible, with high attendance & high dyad satisfaction
					Acceptability	Semi-structured interview (issues affecting ability to participate, satisfaction, application of the problem-solving strategy)	
Relationship Enhancement (RE)	Baucom et al. (2009) [68]	Pilot RCT: RE vs. usual care	14	Female PTs with stage I-II breast cancer & male SPs [USA]	Psychological distress	BSI-18	Compared to usual care, PTs & SPs in RE had improved psychological function & relationship function and PTs had fewer physical symptoms, both immediately post-intervention & one year later
					Post-traumatic growth	PGI	
					Functional QOL	FACT-B (Functional Well-Being subscale)	
					Self image	SIS	
					Relationship function	QMI	
					Sexual function	DISF-SR	
					Fatigue	BFI	
					Pain	BPI	
					Physical symptoms	RSC	
RIPSToP (RelatIonal Psychosexual Treatment for Couples with Prostate Cancer)	Robertson et al. (2016) [69]	RCT: RIPSToP vs. usual care	43	Prostate cancer PTs & SPs [United Kingdom]	Feasibility & acceptability	Recruitment & retention rates; interventionist self-reported fidelity	RE was feasible and acceptable; PTs in RIPSToP had significant improvement in sexual bother compared to those in usual care
					Sexual function	EPIC (Sexual Bother subscale)	
Side by Side	Heinrichs et al. (2012) [70]	RCT: Side by Side vs. Couples Control Program (cancer education control group)	72	Female PTs with stages I-III breast or gynecological cancer ≤4 weeks from diagnosis & male SPs [Germany]	Relationship function	QMI	In Side by Side, PTs had lower fear of progression, and PTs & SPs had decreased avoidance, increased posttraumatic growth, improved communication quality, & more dyadic coping compared to controls
					Communication quality	PFB (Communication subscale)	
					Dyadic coping	DCI	
					Cancer-specific distress	QSC-R23	
					Fear of progression	FPQ	
					Cancer-related avoidance	DII-R (Avoidance-Defense subscale)	
					Post-traumatic growth	PGI	
TSM (Telephone-Based Symptom Management)	Mosher et al. (2016) [71]	RCT: TSM vs. education control condition	106	Lung cancer PTs & family CGs (63% SPs) [USA]	Depression	PHQ-9 (8 item version used)	Compared to education, PTs & CGs in TSM did not have improved depressive symptoms or anxiety, and PTs did not have improved fatigue or breathlessness; TSM also did not improve PT or CG self-efficacy in managing symptoms, nor perceived social constraints from the CG
					Anxiety	GAD-7	
					Pain	BPI-SF	
					Fatigue	FSI	
					Physical symptoms	MSAS (4 items only)	
	Winger et al. (2018) [34]	Secondary analysis of data from Mosher et al. (2016) [71]	51	Subset of PTs from Mosher et al. (2016) [71] Lung cancer PTs ≥3 weeks after diagnosis & family CGs (62.75% SPs) [USA]	Pain	BPI-SF	Assertive communication (taught in TSM) was associated with less PT pain interference & psychological distress; guided imagery (taught in TSM) was associated with less CG psychological distress; however, other coping skills taught in TSM were associated with increases in some PT symptoms (e.g., pain & fatigue interference)
					Fatigue interference	FSI	
					Dyspnea	MSAS (single breathless-ness item)	
					Depression	PHQ-8	
					Anxiety	GAD-7	
TYC (Couple-Based Tibetan Yoga)	Milbury et al. (2015) [72]	Single-arm pilot study	10	PTs with stages I-IIIB NSCLC receiving ≥5 weeks radiation & family CGs (90% SPs) [USA]	Feasibility	Consent rates; session attendance; participant evaluations; questionnaire completion; attrition	Intervention was feasible and most participants perceived the program as useful and beneficial; spiritual QOL improved over time for PTs, and fatigue and anxiety improved over time for CGs
					Psychological distress	CES-D; BSI-18 (Anxiety subscale)	
					Sleep disturbance	PSQI	
					Fatigue	BFI	
					Health-related QOL	MOS-SF-36	
					Spiritual QOL	FACT-Sp	
					Meaning-making	FMCS	
(Unnamed)	Shields et al. (2004) [73]	Non-randomized, controlled trial: 2-session intervention vs. 1-session intervention vs. control	48	Breast cancer PTs & male SPs [USA]	Feasibility & acceptability	Recruitment & retention rates	Intervention was generally feasible & acceptable; 2-session format produced most positive change in psychological well-being and cancer-specific distress
					Psychological well-being	MOS-SF-12 (Mental Health subscale)	
					Cancer-specific distress	IES	
					Relationship function	RDAS	
(Unnamed)	Wagner et al. (2016) [74]	Single-arm pilot study	12	PTs with incurable stage IIB-IV lung or breast cancer & SPs [USA]	Feasibility & acceptability	Recruitment & retention rates	Generally feasible & acceptable; SPs in the intervention had reduced depression and anxiety
					Anxiety & depression	HADS	
(Unnamed)	Mosher et al. (2018) [75]	RCT: peer helping + coping skills vs. coping skills	50	PTs with stage IV GI cancer diagnosed ≥8 weeks prior & family CGs (76% SPs) [USA]	Feasibility	Recruitment, retention, & session completion rates	Intervention was feasible & acceptable; those in the coping skills (control) group had more improvement in meaning in life/peace compared to the peer helping + coping skills intervention
					Acceptability	Purpose-designed scale	
					Spiritual QOL	FACIT-Sp (Meaning/ Peace subscale)	
(Unnamed)	Clark et al. (2013) [76]	RCT: intervention vs. usual care	117	Advanced cancer PTs diagnosed in the past 12 months who were scheduled for radiation therapy & CGs (79% SPs) [USA]	QOL	FACT-G	Intervention PTs had higher overall QOL compared to usual care
	Piderman et al. (2014) [33]	Secondary analysis of data from Clark et al. (2013) [76]	117	See Clark et al. (2013) [76]	QOL	FACT-G; LASA	Intervention PTs had improved spiritual & overall QOL compared to usual care
					Spiritual QOL	FACIT-Sp	
	Lapid et al. (2016) [30]	Secondary analysis of data from Clark et al. (2013) [76]	116	See Clark et al. (2013) [76]	QOL	CQOL-C; LASA	CGs in the intervention (vs. usual care) had improved QOL in several specific domains (including spiritual well-being, mood, vigor/fatigue, and adaptation to cancer), but there were no group differences for overall QOL

Note. Abbreviations that are not defined in the table are listed here, in alphabetical order: ACS = Appraisal of Caregiving Scale; AIS = Appraisal of Illness Scale; APN = advanced practice nurse; BFI = Brief Fatigue Inventory; BHS = Beck Hopelessness Scale; BPI = Brief Pain Inventory; BPI-SF = Brief Pain Inventory-Short Form; BSI-18 = Brief Symptom Inventory-18-item version; CAHS = Cognitive Appraisal of Health Scale; CAS = Constipation Assessment Scale; CBT = cognitive-behavioral therapy; CDS = Caregiver Demands Scale; CES-D = Centers for Epidemiological Studies-Depression; CG = caregiver; COHQOL = City of Hope Quality of Life instruments for patients or caregivers; CQOL-C = Caregiver Quality of Life Index-Cancer; C-SOSI = Calgary Symptoms of Stress Inventory; CSQ = Client Satisfaction Questionnaire; CSSDS = Cancer Support Source Distress Scale; DAS = Dyadic Adjustment Scale; DCI = Dyadic Coping Inventory; DII-R = Dealing with Illness Inventory-Revised; DIS = Dyspnea Intensity Scale; DISF-SR = Derogatis Inventory of Sexual Functioning; DT = Distress Thermometer; EORTC QLQ-C30 = European Organisation for Research and Treatment of Cancer-Quality of Life of Cancer Patients questionnaire; EPIC = Expanded Prostate Cancer Index Composite; EPIC-S = Expanded Prostate Cancer Index Composite-Spouses; FACIT-Sp = Functional Assessment of Chronic Illness Therapy-Spiritual Well-Being Scale; FACT-B = Functional Assessment of Cancer Therapy-Breast; FACT-G = Functional Assessment of Cancer Therapy-General; FACT-HN = Functional Assessment of Cancer Therapy-Head & Neck; FACT-P = Functional Assessment of Cancer Therapy-Prostate; FMCS = Finding Meaning in Cancer Scale; FPQ = Fear of Progression Questionnaire; FRI = Family Relationship Index; FSI = Fatigue Symptom Inventory; GAD-7 = Generalized Anxiety Disorder-7-item scale; HADS = Hospital Anxiety and Depression Scale; HQLI = Hospice Quality-of-Life Index; HWS = Holistic Well-Being Scale; IES = Impact of Events Scale; IES-R = Impact of Events Scale-Revised; IIRS = Illness Intrusiveness Rating Scale; IOS = Inclusion of Other in the Self Scale; ISI = Insomnia Severity Index; LASA = Linear Analog Self-Assessment; LCSES = Lewis Cancer Self-Efficacy Scale; LMISS = Lewis Mutuality and Interpersonal Sensitivity Scale; MAAS = Mindful Attention Awareness Scale; MDASI = MD Anderson Symptom Inventory; MHI = Mental Health Inventory; Mini-MAC = Mini-Mental Adjustment to Cancer Scale; MOS-SF-12 = Medical Outcomes Study-Short Form-12-item version; MOS-SF-36 = Medical Outcomes Study-Short Form-36-item version; MQOL = McGill Quality of Life Inventory; MSAS = Memorial Symptom Assessment Scale; MSIS = Miller Social Intimacy Scale; MUIS = Mishel Uncertainty in Illness Scale; MUIS-B = Mishel Uncertainty in Illness Scale-Brief version; M-VITAS = Missoula Vitas Quality of Life Index; NRS = Numeric Rating Scale; NSCLC = non-small cell lung cancer; ODCNF = Openness to Discuss Cancer in the Nuclear Family scale; OSQ = Omega Screening Questionnaire; PFB = Partnerschaftsfragebogen (Partnership Questionnaire); PGI = Posttraumatic Growth Inventory; PHQ-8 = Patient Health Questionnaire-8-item version; PHQ-9 = Patient Health Questionnaire-9-item version; POMS = Profile of Mood States; POMS-B = Profile of Mood States-Brief; POMS-SF = Profile of Mood States-Short Form; PROMIS = Patient Reported Outcomes Measurement; PSQI = Pittsburgh Sleep Quality Index; PSS = Perceived Stress Scale; PT = patient; QL-SP = Quality of Life Questionnaire for Spouses; QMI = Quality of Marriage Index; QOL = quality of life; QSC-R23 = Questionnaire on Stress in Cancer Patients; RCT = randomized controlled trial; RDAS = Revised Dyadic Adjustment Scale; RN = registered nurse; RSC = Rotterdam Symptom Checklist; SF = short form; SIS = Self-Image Scale; SP = spouse/partner; SPSI-R = Social Problem Solving Inventory-Revised; SW = social worker; USA = United States of America. ^a^ See Appendix A for demographics of each sample.

**Table 2 ijerph-19-13561-t002:** Description of Interventions Included in this Review.

Intervention	Citation(s)	Relevant Pillars of Positive Psychology							Intervention Description		
		PE			E	PR	MM	A	Details of Intervention Delivery	Intervention Components	Additional Information
		M	OH	OG							
**4Cs**	Li et al. (2015) [39]				**🗸**	**🗸**		**🗸**	Six weekly in-person group-based sessions; delivered by a researcher/therapist	Sessions covered broad topics such as primary/secondary stressors, dyadic mediators, dyadic appraisal, and dyadic coping Specific content areas covered include **relationship engagement, caregivers’ feeling of accomplishment, meaning of their role in daily life, relationship with family and friends, maintaining hope when the situation seems hopeless**, and reciprocal self-disclosure	N/A
**ACT**	Mosher et al. (2019) [40]	**🗸**					**🗸**		Six weekly 50-min phone sessions (dyads attended sessions 1 & 4–6 together; sessions 2–3 delivered to PTs & CGs separately); delivered by master’s level SW	Patient and caregiver coping strategies for managing symptoms and distress Experiential practice of **mindfulness** during sessions and at home Practice cognitive defusion and cultivate perspective-taking Identify personal values and **practice values-consistent actions** (p. 635)	The intervention targets processes of the ACT model of behavior change, including mindfulness, perspective taking, cognitive defusion, acceptance, values clarification, and committed action (pp. 634–635).
**CBM**	Milbury et al. (2020) [41]	**🗸**				**🗸**	**🗸**		Four weekly 60-min sessions delivered via FaceTime; delivered by a master’s-level licensed psychological counselor intern	**Mindful meditation on current experiences** and sharing reflections and experiences with the partner **Mindful meditation on interconnectedness and feelings of compassion for partner**, with shared reflections **Gratitude meditation** with mindful/compassionate sharing **Value-based living** (identifying core values & strategies to ensure that lives reflect self-identified values)	Informed by the positive psychology literature and integrates both intrapersonal (i.e., meditations) and interpersonal (i.e., emotional sharing) components
**CDGI**	Saita et al. (2016) [42]	**🗸**				**🗸**	**🗸**		Eight in-person group-based sessions which met every 2–3 weeks for “a couple of hours;” delivered by 2 psychosocial oncology practitioners	Psychoeducation; identify coping strategies, **develop bonds** among group members, introduce dyadic coping **Finding strength and resilience**; integrate illness into broader family history **Relationship as strength and resource; discover positive aspects, resources, and competencies available within close relationships** **Dyads reflect on beauty and strength in spite of illness/treatment**, impact of cancer on intimacy Mind/body connection; focus on **mindfulness**, **relaxation exercise**, handling negative emotions/stress **Making-meaning** (pp. 3–4)	CDGI is a supportive group-based intervention for cancer patient and caregiver dyads theoretically inspired by the Bio-psychosocial Model, the Symbolic Relational Model, and the Psycho-Educational Approach (p. 3).
**CECT**	Collins et al. (2013) [43]; Couper et al. (2015) [44]						**🗸**		Six weekly 60–90 min in-person sessions + 2 follow-up sessions at 10 weeks & 9 months; delivered by mental health professionals supervised by a clinical psychologist and two psychiatrists	CECT aims to address key existential and functional themes including the following (Collins, p. 466): Anxiety about recurrence and death Coping with cancer treatments and their side effects The impact of the diagnosis and treatment on the couple’s relationship, including sexual impact Family concerns, body image and self-image concerns, lifestyle effects and future goals Therapeutic goals of CECT (Couper et al., 2015, p. 37) [44]: Support couples; teach cognitive approach to dealing with anxieties, problem-solving approach to coping **Re-evaluate life’s priorities** as an individual and as couple, **foster authentic living, meaning, purpose** Dealing with grief and losses	CECT combines supportive, existential and cognitive therapy approaches in a structured way to assist couples to **develop a positive attitude, use adaptive coping strategies, and maintain a sense of meaning and authenticity in their lives together** (Couper et al., 2015, p. 36) [44].
**COPE**	McMillan et al. (2006) [45]; McMillan & Small (2007) [32]; Meyers et al. (2011) [46]		**🗸**						Three in-person sessions delivered over 1 month; delivered by trained health educators	**Creativity** (viewing problems from different perspectives to problem-solve) **Optimism (having a positive, but realistic, attitude toward the problem-solving process)**; includes communicating realistic optimism to the patient by showing both understanding and **hope** Goal-setting and developing action steps Expert information (McMillan et al., 2006, p. 217) [45].	COPE addresses problems known to affect patients with cancer including physical symptoms (pain or nausea), psychological symptoms (anxiety or depression), or issues related to resources or relationships, including communicating with one’s health care team or getting support or services from family, friends, and community organizations.
**CSC**	Gremore et al. (2021) [47]					**🗸**			Four 75-min in-person sessions with couples while PT received chemotherapy; delivered by clinical psychologist	Highlighting couples’ **individual and relationship strengths** Learning about problem-solving vs. supportive communication Practicing **supportive communication skills** Identify individual needs and share with partner	Based on social-cognitive processing theory and the intimacy model
**Dignity Therapy**	Wang et al. (2021) [48]			**🗸**			**🗸**	**🗸**	Five to six sessions, including an introductory session, two to three 45–60 min interview sessions, a photo collection and interview transcript editing session, and a session to share a final e-product with the dyad; delivered-by a nurse or physician trained in dignity therapy	Creation of an “e-product” extracted from interviews, photos, and music chosen by the dyad, which could be shared with others Addressed topics such as patients’ life experiences, important roles, most important accomplishments, words or hopes for loved ones, unfinished business, and plans for the future	Based on Confucianism
**DYP**	Milbury, et al. (2018) [49]; Milbury, Li, et al. (2019) [50]; Milbury, Liao, et al. (2019) [51]	**🗸**							12 sessions delivered over course of patient’s radiotherapy, 2–3x per week, 60 min per session; delivered by two certified instructors (International Association of Yoga Therapists)	Joint loosening with **mindfulness** training Asanas with **deep relaxation techniques** Pranayama with sound resonance **Meditation/guided imagery focused on love and compassion for self and caregiver** (p. 333)	With traditional Indian yoga practice in mind, the underlying philosophy of this dyadic intervention was based on principles of interdependence: reciprocal support, teamwork, and equity, which were interwoven in all aspects of the program (p. 333).
**ECG**	Manne et al. (2016) [52]					**🗸**			Eight weekly 90-min in-person group-based sessions; delivered by two therapists (SWs or psychologists) per group	**Focused-breathing relaxation, muscle relaxation, progressive muscle relaxation, guided imagery** Identify and express support needs and **being a good support to one’s partner** **Create “wish list” of positive acts for spouse to do for partner** Constructive communication, stress management and sexual intimacy Problem solving, emotion-focused coping, and partner-assisted cognitive restructuring Preparing couples for survivorship (pp. 5–6)	N/A
**FOCUS**	Northouse et al. (2002, 2005, 2007, 2013, 2014) [53,54,55,56,57]; Harden et al. (2009) [29]; Martinez et al. (2015) [31]; Dockham et al. (2016) [58]; Titler et al. (2017) [59]; Chen et al. (2021) [60]; Titler et al. (2020) [35]		**🗸**			**🗸**			Three monthly 90-min home visits (initial phase) + two monthly 30-min phone sessions (booster phase) after the home visit phase; delivered by master’s-level nurse. Brief version and web-based sessions also exist.	Family involvement (promoting open communication, **encouraging mutual support and teamwork**, **identifying family strengths**, helping children in the family) **Optimistic attitude (practicing optimistic thinking**, sharing fears and negative thoughts, **maintaining hope, staying hopeful in the face of death)** Coping effectiveness (dealing with overwhelming stress, **encouraging healthy coping and lifestyle behaviors**, helping caregivers manage the demands of illness) Uncertainty reduction (obtaining information, learning to be assertive, learning to live with uncertainty) Symptom management (assessing symptoms, **self-care** strategies) (Northouse et al., 2002, p. 1415) [53]	N/A
**I-BMS**	Lau et al. (2020) [37]; Xiu et al. (2020) [38]	**🗸**					**🗸**		Eight weekly 3-h group sessions + 2 follow-up group sessions; first seven sessions, PTs & CGs attended parallel group sessions in different rooms; delivered by two to three facilitators (SW or psychologist);	Psychoeducation about holistic health Mind-body exercises (e.g., Qigong-inspired movement) **Mindfulness-based activities** such as meditation to reduce stress and **cultivating emotional equanimity** Life-review for **reconstructing meanings out of their cancer** (patients) or caregiving (caregivers)	Based on “Daoist philosophy, traditional Chinese medicine (TCM) and Western psychotherapy models…[enables] participants to appreciate the interconnectedness of their bodies, emotions, and spirituality (i.e., sense of peace, meaning), thereby building holistic capacity for transformative changes beyond the reduction of symptoms…I-BMS facilitates well-being through appreciating the interdependence among one’s body, mind and spirit, and building resources for personal growth…” (Lau et al., 2020, p. 391) [37]
**MBSR**	Birnie et al. (2010) [61]	**🗸**							Eight weekly 90-min sessions + one 3- or 6-h weekend silent retreat	Psychoeducation **Mindfulness practices** (including body scan, meditation, awareness of pleasant moments)	N/A
**MBSR-C**	Lengacher et al. (2012) [62]	**🗸**							6-week intervention consisting of three in-person classes (weeks 1, 3, & 6), listening to audiotaped sessions at home on CDs, and at-home practice exercises; delivered by licensed clinical psychologist	Focus on emotional/psychological and physical responses to stressors **Mindfulness practices** (including sitting and walking meditation, body scan, and yoga)	MBSR specifically adapted for cancer context
**MODEL Care**	Cottingham et al. (2019) [36]; Johns et al. (2020) [63]	**🗸**			**🗸**	**🗸**	**🗸**		Six weekly 2-h in-person group sessions + home practice; delivered by facilitator trained in mindfulness practices	Each session had an overall theme, a **mindfulness practice**, didactics, and home practice Session themes included awareness (“meeting ourselves where we are in honesty and kindness”); perception and creative responding; relational presence; and **mindful dialogue**	Draws on MBSR and mindful speaking/listening practices
**PIP/MPI**	Chien et al. (2020) [64]			**🗸**					PIP: Information manuals & 6 weeks of telephone counseling; MPI: Weekly psychosocial information film, psychosocial information manual & professional support for 6 weeks. Both delivered by trained nurses.	Education about prostate cancer, sexual function and management Emotional adjustment and **maintaining positive emotion** Coping/stress management Diet and physical activity in context of cancer Social resources	Based on transactional model of stress and coping
**PERC**	Song et al. (2015) [65]		**🗸**						Two mandatory + five optional web-based sessions over up to 8 weeks (dyads could complete together or separately); self-guided	Modules from the FOCUS program (described above) that explore family involvement, **optimistic attitude**, coping effectiveness, uncertainty reduction, and symptom management Psychoeducation	“PERC takes a supportive educational approach to helping couples work together to mitigate the impact of patients’ symptoms after treatment for prostate cancer…The mandatory modules provided information about how couples can work as a team (e.g., communication) and various survivorship issues (e.g., distress, relaxation, communication with healthcare team). The optional modules focused on the management of prostate cancer-specific and general symptoms” (p. 184).
**PICP**	Kayser et al. (2010) [66]					**🗸**			Nine 60-min in-person sessions, once every 2 weeks; delivered by a SW	Assessment of the couple’s relationship and social support network Integrate tasks of Illness into a couples daily routine Personal coping and preserving physical and psychological health, learning new coping skills Enhance the couple’s communication and **promote supportive exchanges** Enhance intimacy and sexual functioning (p. 25)	PICP developed using a cognitive-behavioral framework. Sessions (left) organized to go from “less personal and emotional issues to more intimate and emotion-focused issues” (p. 24).
**PSE**	Bevans et al. (2010) [67]		**🗸**						Four in-person sessions (median = 45 min) over course of PT’s hematopoietic stem cell transplant (pre-transplant to 4 weeks post-discharge); delivered by clinicians “with advanced degrees” (e.g., SW, nurse specialist)	COPE: **creativity**, **optimism**, planning, expert information Home Care Guide for outline plans for common cancer problems (p. 4)	Seeks “to **empower dyads** to cope with cancer and cancer treatments using two major processes from the social problem-solving literature: Problem orientation and problem-solving skills. **An optimistic approach to managing the problem and permission to be creative was reinforced throughout the session**” (p. 4). [USA]
**RE**	Baucom et al. (2009) [68]			**🗸**			**🗸**		Six 75-min in-person sessions, once every 2 weeks; delivered by psychology doctoral students	Breast cancer education Communication skills for decision-making and sharing thoughts/feelings regarding cancer-related issues Approaching breast cancer as a couple; promoting a healthy sexual adaptation and body image **Maintaining positives in life during difficult times** Finding benefits and meaning in life in the face of breast cancer.	N/A
**RIPSToP**	Robertson et al. (2016) [69]					**🗸**			Six 50-min in-person sessions, once every 2–3 weeks; delivered by registered therapy practitioners	**Couple’s communication style and relationship (how they convey love, support, understanding, companionship, affection)** Patterns of illness, coping, and affection (**family resilience**, dyadic adjustment, family roles) Couple intimacy before/after cancer (psychoeducational approach to promote closeness/intimacy) (p. 1236)	Included “assistance with emotional disclosure, psychoeducation, relational and sexual needs, and dyadic adjustment and coping” (p. 1234).
**Side by Side**	Heinrichs et al. (2012) [70]					**🗸**			Four 120-min home visits, once every 2 weeks; delivered by therapists	Individual and relationship skills for partners Centers on communication skills (train couples in speaker and listener guidelines) and **positive forms of dyadic coping** training (p. 244).	Origins in CanCOPE, a couple-based coping intervention. “Significant emphasis on sharing thoughts and feelings and couple’s communication in cancer-related areas” (p. 243). “Within the framework of a cognitive-behavioral theory of conceptualizing relationship difficulties as well as **building on couples’ strengths**, we based our approach and hypotheses on an adaptation model of couples functioning” (p. 240).
**TSM**	Mosher et al. (2016) [71]; Winger et al. (2018) [34]	**🗸**		**🗸**					Four weekly 45-min phone sessions; delivered by a SW	**Mindfulness exercise**, guided imagery, pursed lips breathing Cope with distressing thoughts based on the type of thought, including cognitive restructuring, problem solving, and self-soothing/emotion-focused strategies Assertive communication Schedule **pleasant activities**, pacing, and coping skills practice (Winger et al., 2018, p. 1343) [34].	“The primary goal of the intervention was to teach patients and caregivers various evidence-based cognitive-behavioral and emotion-focused strategies for managing anxiety and depressive symptoms, pain, fatigue, and breathlessness.” (Mosher et al., 2016, p. 471) [71].
**TYC**	Milbury et al. (2015) [72]	**🗸**							10–15 45–60 min in-person sessions (2–3 weekly sessions over 5–6 weeks, delivered alongside radiation treatments); interventionists not described	Deep breathing awareness with visualization Breath retention exercises (e.g., 4-Part Breath) **Mindfulness and focused attention through guided meditation** A brief **compassion-based meditation** (p. 2)	Starting with session 1, instructors convey that the practice targets the needs of both dyad members with a focus on their interconnectedness. Starting with session 5, the dyad is given time for expressing emotional attachment, closeness, and **compassion** (e.g., holding hands, gazing into each other’s eyes, **verbal sharing of love and affection**) (pp. 2–3).
**Unnamed**	Shields et al. (2004) [73]						**🗸**		Two-session intervention: two 4-h in-person group-based sessions; interventionists not described one-session intervention: one 4-h in-person group-based session; interventionists not described	Compare and contrast patients’ and spouses’ experiences with cancer Strengthen couples’ communication about emotion **Find meaning and perspective** (couples make a timeline of their life together) (p. 100)	“Our workshop builds on established family oriented interventions for medical illness, techniques developed for marital therapy, and cognitive therapy techniques adapted for use with couples” (p. 100).
**Unnamed**	Wagner et al. (2016) [74]		**🗸**				**🗸**		Four 60-min in-person sessions; delivered by psychologist	**Meaning in life** (life review) **Hopes for the future** (determine each partner’s values and wishes for end-of-life approaches) **Social connectedness** (recall moments in life that felt **meaningful**, reflect on how cancer affected their **sense of meaning**) (pp. 548–549)	“Grounded in existential psychotherapy and designed to **increase meaning in life** and sense of transcendence, **determine wishes and hopes**, and help patients and their partners communicate more openly about death and dying” (p. 548).
**Unnamed**	Mosher et al. (2018) [75]			**🗸**		**🗸**			Five weekly 50 to 60-min phone sessions; delivered by psychology doctoral students supervised by psychologists	Manage physical symptoms (coping skills for pain management/fatigue, relaxation); self-care habits Manage stress (coping skills for stress management through **pleasurable activities**) **Maintain relationships** (coping skills for dealing with negative reactions from others and loneliness)	N/A
**Unnamed**	Clark et al. (2013) [76]; Piderman et al. (2014) [33]; Lapid et al. (2016) [30]		**🗸**	**🗸**			**🗸**		Six 90-min in-person sessions, three times per week (CGs attended two per week) + 10 phone sessions, once every 2 weeks; in-person sessions delivered by a psychologist supported by other staff (e.g., APN, chaplain, SW), phone sessions, led by a psychologist or physical therapist	Conditioning exercises, education, cognitive behavioral strategies for coping with cancer, open discussion and support, and a deep breathing or guided imagery **relaxation** segment (Clark et al., 2013, p. 5) [76] Health behavior changes, benefits of physical activity, and tracking symptoms **Self-care**, symptom management, and treatment education topics **Spirituality (life review; meaning and purpose**; grief, loss, **hope**, **and blessings**) Coping with cancer, problem solving skills, **relaxation** training, mood management Social needs (advanced directives, finances, community resources) Defining your QOL (Lapid et al., 2016, p. 1402) [30]	“The structured, multidisciplinary intervention focused on specific strategies to address all five QOL domains. The content was developed by a multidisciplinary treatment and designed to impact physical, mental, social, emotional, and spiritual QOL” (Clark et al., 2013, p. 4) [76].

Note. MM = meaning-making, PE = enhancing positive emotions (M = mindfulness, OH = optimism/hope, OG = other/general positive emotions), E = engagement, PR = positive relationships, A = accomplishment. Words/phrases in bold represent components of the intervention that are components of positive psychology interventions. See Results section for more information. Citations refer to References section of main text.

## Data Availability

Not applicable.

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
