# Peer review of "Positive Psychology Approaches to Interventions for Cancer Dyads: A Scoping Review"

_ijerph, 2022, doi:10.3390/ijerph192013561_

Round 1

Author Response

Methods

  1. I did not understand why you included studies only after 2002. Timeframe of inclusion is not justified – provide a brief explanation.

AUTHORS’ RESPONSE: We did not have an exclusion criterion related to date of publication; 2002 was simply the earliest year of publication among our final set of included articles. We apologize for the confusion and now state this explicitly in the Materials and Methods section (under section 2.3 Eligibility Criteria; p. 4, line 143). We also modified the language in the first sentence of the Results section to provide additional clarity.

Results

  1. I much prefer Supplemental Table 4 to the Table 1 included in the manuscript. It is a much better synthesis of your analysis. Consider further consolidating or publishing the supplemental table instead.

AUTHORS’ RESPONSE: Per this reviewer’s suggestion, we have moved Supplemental Table 4 out of the Supplementary Materials and into the main manuscript document (now labeled Table 2). (All tables in both the manuscript and Supplementary Materials have been renumbered accordingly.)

  1. Minor detail – sometimes page numbers are included, but not consistently.

AUTHORS’ RESPONSE: We apologize for this oversight. Our resubmission now includes page numbers in the top right corner of each page.

  1. Outcomes of interventions are referred to in the discussion, but not summarized in the results. The following sentence could be moved from the discussion and expanded. “The primary outcomes assessed in these interventions were individual psychological well-being and quality of life or distress (including anxiety and depression), dyadic coping or adjustment, and, less often, physical symptoms such as fatigue or pain.”

AUTHORS’ RESPONSE: We now include two additional columns in Table 1 summarizing the primary outcomes assessed in each study: one column for the construct, and one column for the measure used to assess that construct. We have also added a short summary of this information to the Results section (p. XX, line XX) and again in the Discussion section (p. XX, line XX).

  1. Based on the variation in sample sizes and design, the authors allude to variation in study quality but stop short of critique. Adding a quality assessment or even summarizing the number of studies that used CONSORT or other reporting guidelines would help readers compare quality across studies.

AUTHORS’ RESPONSE: We appreciate this suggestion, although it is still difficult to compare study quality in this way given the variety in study designs—particularly as non-randomized or single-arm trials have no standard reporting guidelines like CONSORT to follow (https://www.equator-network.org/). Additionally, a formal quality assessment is not a requirement for scoping reviews (see: Peters MD, Godfrey CM, Khalil H, McInerney P, Parker D, Soares CB. Guidance for conducting systematic scoping reviews. Int J Evid Based Healthc. 2015;13(3):141-146.) As the reviewer noticed, however, we did try to allude to quality by noting the number of studies that were randomized controlled trials (RCTs) vs. single-arm trials (vs. secondary analyses) in the Results section (p. 4), as well as the variety of sample sizes used across studies.

  1. Consider adding a description of the racial/ethnic and income diversity of samples. I think most of these studies recruited mostly White, middle-class samples.

AUTHORS’ RESPONSE: This reviewer is correct that most of these studies utilized White, relatively high-socioeconomic status samples. We have added a new Supplemental Table 4* that includes information about key demographic variables for the samples in the included studies: race/ethnicity, age, gender, income, and education. We have also added a comment about diversity of samples in section 4.1 (Limitations and Future Directions; p. 7, line 317)--also noted below in response to this reviewer’s comment #7. (*Note that the old Supplemental Table 4 was moved to the main text and is now Table 2; see response to this reviewer’s second comment above.)

 Limitations

  1. Address generalizability based on limited diversity of samples. Who does this body of evidence represent?

AUTHORS’ RESPONSE: We have added a comment on the diversity of samples in the section 4.1 (Limitations and Future Directions; p. 7, line 317).

Other Comments:

  1. I really appreciated the discussion of ‘Dyadic’ work. This was nicely addressed in the discussion. I wonder if something should be said earlier when you are defining inclusion? Perhaps with this sentence, We stipulated that the intervention must be delivered to both members of the dyad together for at least part of the intervention.”

AUTHORS’ RESPONSE: This is a great suggestion! We have added some additional text discussing our operationalization of “dyadic” (and referring readers to the Discussion section for additional commentary on this) in section 2.3 (Eligibility Criteria; pp. 3-4, line 140).

  1. deeper well of resilience to draw from”…really lovely imagery

AUTHORS’ RESPONSE: Thank you, we appreciate the positive feedback!

Minor Suggested Edits:

  1. 20 of these 28 interventions (71.4%) were reported on in only a single included publication.”
  2. “The FOCUS intervention was published on by far the most frequently published 11 studies…” This paragraph has a better flow with the first paragraph of results. Suggest switching this and the 2nd paragraph which begins, “There was substantial variation in how the non-patient dyad member was defined.”

 “COPE (n=xx) and DYP (n=xx) were the only other interventions that published more than two papers.”

AUTHORS’ RESPONSE: Thank you for these suggested edits. We have clarified the writing in these places.

Reviewer 2 Report

1. Why was the positive psychology approach chosen in this manuscript from the Dyads cancer intervention? What are the main scientific questions addressed by this study? These issues require clearer clarification.

2. Theoretical analysis is still simple to fully reflect the inherent logical relationship between the Dyads cancer intervention and the factors that influence it. Does the Dyads cancer problem in the case area have any salient characteristics? Is the problem representative? These problems require clearer clarification in the method.

3. A more in-depth analysis of the mechanisms of positive psychology approaches influencing Dyads cancer intervention is needed in the analysis of the results and theoretical implications of the theoretical value of this study, and a more in-depth analysis and targeted analysis of the practical implications is needed to provide a better reference or guide for the case area.

4. Some of the limitations of the analysis of the study area, the logical theoretical relationship of positive psychology and Dyads cancer, the basis of the plots and selected samples, and targeted suggestions should be illustrated more clearly.

Author Response

  1. Why was the positive psychology approach chosen in this manuscript from the Dyads cancer intervention? What are the main scientific questions addressed by this study? These issues require clearer clarification.

AUTHORS’ RESPONSE: We apologize if we are misunderstanding this reviewer’s questions, but as we describe in the second paragraph of the introduction, there is no one positive psychology approach (PPA); rather, we aimed to provide a review of different interventions that are based on or incorporate the 5 pillars of positive psychology (i.e., enhancing positive emotions, engagement, positive relationships, meaning-making, and accomplishment). We did not conduct a dyadic cancer intervention ourselves. As noted in the last paragraph of the introduction, the objective of this scoping review was to conduct a thorough search of the literature to provide an overview of the available research evidence for use of a dyadic PPA with cancer patients and their caregivers.

  1. Theoretical analysis is still simple to fully reflect the inherent logical relationship between the Dyads cancer intervention and the factors that influence it. Does the Dyads cancer problem in the case area have any salient characteristics? Is the problem representative? These problems require clearer clarification in the method.

AUTHORS’ RESPONSE: We apologize, but we had some trouble understanding this feedback; we have tried our best to respond based on our understanding of what this reviewer is asking: We include cancer characteristics (site) in the Results section (p. 4, line 170), but given the small numbers of each cancer type reported in the included studies—if information about cancer site or stage is reported at all—it is difficult to draw firm conclusions about the representativeness of these samples with regard to cancer characteristics. The only claims we can make about a representative problem is that cancer caregiving can be challenging regardless of site, and interventions are needed to support the health and well-being of both caregivers and cancer patients.

  1. A more in-depth analysis of the mechanisms of positive psychology approaches influencing Dyads cancer intervention is needed in the analysis of the results and theoretical implications of the theoretical value of this study, and a more in-depth analysis and targeted analysis of the practical implications is needed to provide a better reference or guide for the case area.

AUTHORS’ RESPONSE: We agree that a more thorough assessment of the mechanisms of these positive psychology approaches is needed in the future! However, the goal of the current paper was to thoroughly survey the existing interventions using positive psychology approaches (PPAs) for dyads coping with cancer. Therefore, an in-depth analysis of mechanisms is outside the scope of the current paper. However, we include this as a future direction in the Discussion section: in the first sentence of section 4.2 (Clinical Implications; p. 8, line 350), and in the last sentence of section 5 (Conclusions; p. 9, line 376). Practical implications for these existing dyadic PPAs are also described in section 4.2 (Clinical Implications; p. 8, line 349).

  1. Some of the limitations of the analysis of the study area, the logical theoretical relationship of positive psychology and Dyads cancer, the basis of the plots and selected samples, and targeted suggestions should be illustrated more clearly.

AUTHORS’ RESPONSE: We apologize, but we are having difficulty understanding what the reviewer means here. If the reviewer would be so kind as to clarify or expand upon this comment, we would be more than happy to address their concerns. However, without some clarification from the reviewer or editor, we are unsure as to what specifically the reviewer would like to see addressed with regard to limitations.